# Genetic Basis of Pigment Dispersion Syndrome and Pigmentary Glaucoma: An Update and Functional Insights

**DOI:** 10.3390/genes15020142

**Published:** 2024-01-23

**Authors:** Shisong Rong, Xinting Yu, Janey L. Wiggs

**Affiliations:** 1Ocular Genomics Institute, Department of Ophthalmology, Massachusetts Eye and Ear, Mass General Brigham, Harvard Medical School, Boston, MA 02114, USA; janey_wiggs@meei.harvard.edu; 2Department of Medicine, Brigham and Women’s Hospital, Mass General Brigham, Harvard Medical School, Boston, MA 02115, USA; xinting.yu@bwh.harvard.edu; 3Broad Institute of Harvard and MIT, Cambridge, MA 02142, USA

**Keywords:** pigment dispersion syndrome, pigmentary glaucoma, genome-wide association study, genetic associations, exome sequencing, systematic review, animal models

## Abstract

Pigment Dispersion Syndrome (PDS) and Pigmentary Glaucoma (PG) comprise a spectrum of ocular disorders characterized by iris pigment dispersion and trabecular meshwork changes, resulting in increased intraocular pressure and potential glaucomatous optic neuropathy. This review summarizes recent progress in PDS/PG genetics including rare pathogenic protein coding alterations (*PMEL*) and susceptibility loci identified from genome-wide association studies (*GSAP* and *GRM5/TYR).* Areas for future research are also identified, especially the development of efficient model systems. While substantial strides have been made in understanding the genetics of PDS/PG, our review identifies key gaps and outlines the future directions necessary for further advancing this important field of ocular genetics.

## 1. Introduction

Pigment Dispersion Syndrome (PDS) is characterized by the spread of a pigment from the iris pigment epithelium and iris pigment accumulation on the anterior segment structures, including the cornea and the trabecular meshwork [1]. Iris transillumination defects (ITD), Krukenberg’s spindle, pigmented trabecular meshwork (TM), and retinal lattice degeneration of the retina are among the important manifestations in patients with PDS. In some cases, the accumulated pigment may lead to TM alterations and thus interfere with aqueous humor outflow, causing an increase in IOP and the development of glaucomatous optic neuropathy, which leads to the diagnosis of pigmentary glaucoma (PG).

Pigment Dispersion Syndrome has a prevalence of approximately 2.5% in European Caucasian populations [2] with lower prevalence in African [3] and Asian [4,5] populations. The demographic features suggested as risk factors for PDS includes male gender [6,7], Caucasian ethnicity, familial history [8], and myopia [7,9,10,11]. Pigmentary Glaucoma, a form of secondary open-angle glaucoma, impacts around 1.4 individuals per 100,000 annually, emerging as a notable cause of visual impairment and blindness, predominantly in younger adults [1,4,7,12]. Pigment dispersion syndrome and PG are recognized as part of a continuous spectrum of the same pathological condition. Studies indicate that 10% to 50% of individuals with PDS will eventually progress to PG [1,7,12,13,14]. In the Western world, PG constitutes approximately 1–1.5% of all diagnosed glaucoma cases [15]. However, a significant proportion of PDS cases remain undetected. This underdetection could be attributed to the typical onset of PDS during the third or fourth decade of life, and the dense pigmentation of the iris in non-Caucasian individuals may conceal subtle ITD, leading to an underestimated incidence and prevalence [1,7,12,13]. Beyond PG, PDS also increases the risk of PDS-associated retinal detachment, with occurrence rates ranging between 6% and 7%, regardless of glaucoma presence [12]. Hence, a comprehensive understanding of the disease mechanisms, coupled with early diagnosis and personalized treatment strategies for PDS/PG, is critical to mitigate disease progression. Despite this need, the precise pathophysiological mechanisms underlying PDS remain incompletely elucidated, and as such, targeted treatments for PDS have not yet been developed.

Pigment dispersion syndrome and PG are inherited eye conditions. Their inheritance is strongly supported by compelling evidence, including familial transmission studies [10,16,17], heritability research [8,18], and genetic association studies [19]. In some families, an autosomal-dominant inheritance pattern has been suggested [12,17,20]. Understanding the genetic underpinnings of PDS/PG is crucial for several reasons. First, insights into the genetic basis of PDS/PG open avenues for novel therapeutic targets. Second, it facilitates the development of precise diagnostic tools, enabling early detection of at-risk individuals through genetic screening. Third, unraveling genetic associations aids in the stratification of patients based on their genetic risk, allowing for personalized management strategies [21].

Studies on the genetic landscape of PDS/PG have revealed a complex and multifactorial genetic basis. While pedigree-based analysis can suggest autosomal-dominant inheritance [12,17,20,22,23], this is not universally observed. Genetic linkage analyses [11,20,22], recent genome-wide association analysis, and other genetic studies [19,24,25,26] have identified several genetic loci and polymorphisms associated with PDS/PG. Despite these advances, the pathogenic mechanisms underlying PDS/PG remains largely unknown.

This review aims to systematically collate and analyze the existing literature on the genetic basis of PDS/PG. We critically evaluated the genetic studies conducted to date, assess their methodologies, findings, and limitations, and explored the functional and translational implications of these genetic insights in the management of PDS/PG.

## 2. Identification of PDS/PG Genetic Studies

We performed the literature search using Boolean logic and the search terms with controlled vocabularies (i.e., Medical Subject Heading terms) in the PubMed/MEDLINE databases. The search terms were constructed as follows: (pigment dispersion syndrome OR pigmentary glaucoma) AND (Medical Genetics OR genotype OR genetics [Subheading] AND genetics) (Appendix A). Additionally, we searched the Online Mendelian Inheritance in Man (OMIM) [27] and GWAS Catalog [28] databases. The search terms used in OMIM and GAWS Catalog were “pigment dispersion syndrome” OR “pigmentary glaucoma”. We summarized all the records that met the following criteria: (1) study tested associations between genetic variants and PDS or PG; (2) study population was clearly defined; and (3) diagnosis of PDS or PG was based on clinical data. We also scanned the reference lists of the research articles, editorials, or reviews identified during the screening process to include all relevant publications. The last search was done on 28 November 2023.

The literature search yielded 111 records. Among them, 73 relevant reports were selected for full-text review. The resulting 61 reports were summarized (Figure 1).

## 3. Genetic Factors Identified in PDS/PG

### 3.1. Genetic Loci Linked to PDS/PG

Advances in genetic research have elucidated the complex and heterogeneous genetic basis of PDS/PG. Early descriptions of affected pedigrees suggested autosomal dominant inheritance, and genetic linkage analysis of a large pedigree with suggested dominant inheritance identified 7q35-q36 (GPDS1, glaucoma-related pigment dispersion syndrome 1, OMIM 600510) as a region harboring a causal gene [20]. However, a causal gene located within this region has not been identified.

Genetic investigation has mapped another region on chromosome 18 associated with PDS/PG. Analysis of four additional pedigrees, which were not linked to GPDS1, revealed significant linkage to the 18q21 region, consistent with an autosomal-dominant inheritance pattern [29]. A large-scale chromosomal deletion was identified in a single case of an Estonian male with PDS affecting 18q22.1; however, this case also harbored a deletion at 2q22.1 [30]. 

Table 1 summarized the above-mentioned genetic linkage studies.

### 3.2. Candidate Gene Studies

In the quest to elucidate the genetic underpinnings of PDS/PG, a series of candidate gene studies have provided mixed insights. Lynch et al. explored the association between the human *TYRP1* gene and PG, inspired by its role in the DBA/2J mouse model [31]. However, causal variants were not identified. Jaksic et al. reported a case of PG coinciding with central retinal vein occlusion in a patient with the *MTHFR* C677T homozygous variant, highlighting hyperhomocysteinemia as a potential risk factor; although, this finding has not been replicated in larger studies [32].

In a broader context, Fingert et al. investigated the *TBK1* gene, previously implicated in normal-tension glaucoma, across different open-angle glaucoma subtypes including PG. The absence of *TBK1* copy-number variations in PG patients suggested a distinct genetic basis from other glaucoma types [33]. 

Studies exploring the association between the *LOXL1* gene polymorphisms and PDS/PG have yielded inconclusive results mainly due to insufficient statistical power in each study, necessitating a comprehensive meta-analysis. Giardina et al. identified significant allele associations in Caucasian patients with PDS/PG for SNP rs2304722, suggesting that certain *LOXL1* haplotypes may influence PDS/PG [34]. Contrastingly, Wolf et al. found no major influence of *LOXL1* polymorphisms on the pathophysiology of PG in a German cohort; although, a nonsynonymous polymorphism might predict age at disease onset [35]. Additionally, Rao et al. reported no significant association between the common *LOXL1* SNPs and PDS/PG in their study group, further indicating the specificity of these SNPs to exfoliation syndrome and glaucoma [36].

### 3.3. Genome-Wide Association Study

A genome-wide association study (GWAS) published in 2022 marks a notable advancement in understanding the genetic factors contributing to PDS/PG [19]. Using 574 cases with PG or PDS and 52,627 controls of European descent, two novel loci were significantly associated with PDS/PG [19]. The gamma secretase activator protein (*GSAP*) gene was identified with the lead SNP rs9641220 (*p* = 6.0 × 10^−10^). This gene plays a role in Alzheimer’s disease and is involved in the pigmentation processes. The glutamate metabotropic receptor 5 (*GRM5*) gene, in linkage disequilibrium with the tyrosinase (*TYR*) gene, was identified with the lead SNP rs661177 (*p* = 3.9 × 10^−9^). This locus is associated with ocular pigmentation and possibly with retinal detachment and pigmentation traits in hair and skin. The identified SNPs at *GSAP* and *GRM5/TYR* loci explain approximately 6.9% of the heritable PG risk. Appendix A summarized the candidate gene studies and the genome-wide association study of PDS/PG.

### 3.4. Genetic Factors Identified by Exome Sequencing Studies

#### 3.4.1. Identification of *PMEL* Variants in PDS/PG

Using whole-exome sequencing of two independent pedigrees with suggestive autosomal-dominant inheritance, Lahola-Chomiak et al. identified *PMEL* nonsynonymous variants segregating with the disease. Their research demonstrated that *PMEL*, vital in melanosome function for melanin synthesis, storage, and transport, when mutated, leads to defective amyloid fibril formation and pseudomelanosome structural changes. These results suggested that *PMEL* variants act as dominant negative mutations disrupting normal ocular pigmentation and function. Importantly, their study utilized CRISPR-Cas9-induced *pmela* mutations in zebrafish, manifesting in ocular pigmentation defects and abnormal anterior segment of the eye, underscoring the zebrafish model’s potential in studying these mechanisms [26].

Further advancing this understanding, Hodges et al. focused on *PMEL*’s repeat domain, pivotal in functional amyloid formation in the eye. They engineered a CRISPR-induced mutation in zebrafish *PMEL*, replicating the dysamyloidosis pathomechanism seen in human PG. This mutation, subtly altering *PMEL*’s repetitive region, led to disrupted eye morphogenesis and pigment deposition in retinal melanosomes in zebrafish. While the study did not observe glaucomatous changes in IOP or retinal morphology in the mutants, it supported the crucial role of *PMEL*’s repeat domain in ocular pigment pathology. This aligns with the dominant inheritance pattern observed in human PG and further validates the zebrafish as a promising model for exploring *PMEL*-related dysamyloidosis and its implications in pigment dispersion pathologies [37].

#### 3.4.2. Exome Sequencing-Based Survey of Relevant Genes in Large PDS Cohorts

In the comprehensive study by van der Heide et al., the genetic underpinnings of PDS/PG were investigated through a tiered exome analysis in a cohort comprising 198 PDS patients and various control groups [25]. The primary analysis focused on five melanosome-related genes (*TYRP1*, *GPNMB*, *LYST*, *DCT*, and *MITF*), previously implicated in pigment dispersion in mice; however, the enrichment of loss-of-function mutations was not observed in these genes among PDS patients. A broader secondary analysis evaluated an additional 21 genes potentially linked to PDS and identified rare variants in *MC1R*, *SLC45A2*, and *TYR*, but these too lacked a statistically significant association with PDS. Intriguingly, a genome-wide scan highlighted four unique non-silent mutations in the *MRAP* gene exclusively in PDS patients, suggesting a potential link to PDS. However, the subsequent analysis in an expanded cohort (415 cases and 1645 controls) in the same study did not corroborate a robust association. Immunohistochemical analysis revealed abundant MRAP protein in the iris and other ocular tissues involved in PDS, hinting at a possible pathophysiological role. The study concludes that while certain rare mutations were identified, none displayed a clear and statistically significant correlation with PDS, suggesting a complex genetic landscape for PDS/PG that might not be fully discernible through exome analysis alone. This underscores the necessity for further research, potentially integrating larger cohorts and functional studies to unravel the intricate genetic mechanisms underlying these ocular conditions.

#### 3.4.3. *CPAMD8* (C3 and PZP Like Alpha-2-Macroglobulin Domain Containing 8)

Another study explored PDS/PG genetic factors in two Chinese pedigrees with suggested autosomal recessive inheritance and 38 sporadic PG patients using whole-exome sequencing and Sanger sequencing [24]. The study identified compound heterozygous variants in the *CPAMD8* gene in five patients with typical clinical phenotypes of PDS/PG from the pedigrees. One mutation, c.1015G>A (p.V339M), was also found in one of the 38 sporadic patients with PG. Additionally, 13 *CPAMD8* heterozygotes variants were identified in the sporadic PG cases, possibly providing further support for *CPAMD8*’s involvement in PDS/PG. Bioinformatics analyses classified these variants as damaging, destabilizing, or deleterious, suggesting a functional impact on the protein. This study supports autosomal recessive inheritance of PDS/PG in some families and highlights the potential role of *CPAMD8* variants in ocular phenotypes beyond the previously established links with anterior segment dysgenesis [38,39,40] and other glaucoma subtypes (i.e., primary open-angle glaucoma (POAG), primary angle-closure glaucoma (PACG), and congenital glaucoma) [39,41,42,43]. This research underscores the complexity of genetic factors in PDS/PG and opens new avenues for understanding PDS/PG genetic components.

Table 2 summarized the above-mentioned exome-sequencing studies.

## 4. Shared Genetic Factors between PDS/PG and Other Conditions and Traits

The study of shared risk factors between PDS/PG and other conditions and traits can provide new insights into the underlying disease complexity as well as pathogenesis. 

Despite clinical similarities, genetic studies refute a shared etiology between PDS/PG and juvenile open-angle glaucoma, with no linkage to the 1q21-q31 chromosome region [11,22]. The role of *MYOC* mutations in PDS/PG remains inconsistent and limited [44,45,46,47,48,49]. No direct causal link has been established between *LOXL1* variants in pseudoexfoliation syndrome and PDS/PG, necessitating further research [34,35,36,50]. Additionally, *TBK1* gene variations are not associated with PG [33].

Emerging evidence suggests a genetic connection between myopia and PDS/PG, potentially influenced by eye size [18,19], and an inverse correlation between iris pigmentation genes and PG. Hyperhomocysteinemia, particularly in the context of the *MTHFR* C677T mutation, is highlighted as a risk factor for ocular pathologies [32]. While PDS/PG features in syndromic diseases like Knobloch syndrome, linked to *COL18A1* mutations, and potentially in Marfan syndrome, conclusive genetic causality is yet to be established [51,52,53,54,55].

In summary, current research does not support a shared genetic basis between PDS/PG and these ocular and systemic conditions.

## 5. Functional Insights into PDS/PG Pathogenesis Using Animal Models

### 5.1. Mouse Model

The investigation into the genetic basis of PDS/PG has been significantly advanced by animal models, particularly the DBA/2J mouse. This model has been instrumental in both PDS/PG research [56,57,58,59,60,61,62,63,64,65,66,67] and in broadening our understanding of glaucomatous optic neuropathy [67,68,69,70,71]. DBA/2J mice exhibit a set of traits—iris atrophy, pigment dispersion, elevated IOP, and a glaucoma-like degeneration of retinal ganglion cells—akin to human PDS/PG [67]. Subsequent research identified two primary genes, *Tryp1* and *Gpnmb*, as responsible for these traits [68].

The involvement of melanosome genes in iris pigment dispersion pathogenesis is further evidenced by observations in other mouse models. Notably, mutations in genes involved in melanin synthesis have been linked to iris pigment dispersion and atrophy. This is exemplified in the LT/SvEiJ inbred mouse, where the Tyrp1 b-lt allele, characterized by a single missense mutation in Tyrp1, is associated with iris pigment dispersion [72,73]. Similarly, the Tyrp1 b allele, with two missense mutations, leads to iris atrophy in both DBA/2J and YBR/EiJ inbred mouse strains [56,68,74]. The nm2798 spontaneous coat color variant, arising from the dopachrome tautomerase (Dct) allele Dct slt-lt3J, is also implicated in iris pigment dispersion [72].

In a comprehensive genetic analysis targeting the ITD in DBA/2J mice, the oculocutaneous albinism type 2 (Oca2) gene emerged as a significant regulator. Oca2 plays a crucial role in melanin synthesis via melanosomal pH control [75]. Its human counterpart, OCA2, is known to influence iris pigmentation, being the causative gene for both oculocutaneous albinism type 2 (OMIM number 203200) and variations in iris color [76,77]. This discovery bridges a vital link between mouse model findings and human ocular genetics, underscoring the translational potential of these animal studies in understanding and potentially treating PDS/PG.

In the study of iris pigment dispersion, the role of genes associated with melanosome function, but not directly with melanin synthesis, has been increasingly recognized. Notably, the DBA/2J mouse strain exhibits iris pigment dispersion attributed to a C-terminally truncated allele of the glycoprotein nonmetastatic melanoma protein b (Gpnmb), Gpnmb^R150X^ [68]. Gpnmb, though not directly involved in melanin synthesis, plays a critical role in maintaining melanosome structural integrity and sequestering cytotoxic melanin synthesis intermediates [68]. Interestingly, a similar C-terminal truncation in the Pmel gene (Si allele) leads to generalized melanocyte dysfunction and pigmentary abnormalities highlighting the gene-specific impact on melanosome function [78,79,80]. Gpnmb’s additional roles in neuronal- and immune-cell adhesion are also crucial in the glaucoma pathology seen in DBA/2J mice, suggesting that understanding the immune implications of Gpnmb^R150X^-mediated iris pigment dispersion could shed light on the immune system’s role in PDS [57,81].

Another gene of interest, *Lyst*, encodes the lysosomal trafficking regulator protein, crucial for delivering components to early stage melanosomes. Mutations in Lyst can lead to Chediak–Higashi syndrome (OMIM: 214500) [82]. The Lystbg-j allele, specifically, results in a beige coat color in C57BL/6J mice, along with pronounced pigment dispersion and increased melanosome volume, sharing phenotypic traits with both PDS and pseudoexfoliation syndrome [72,83]. The exact molecular mechanisms behind these phenotypes, potentially linked to cytotoxic melanin synthesis intermediates, remain to be elucidated.

Additionally, a comprehensive genetic analysis identified several genes not directly involved in melanin synthesis but influencing ITD. These include the motor protein Myosin Va (Myo5a), the signaling protein kinase C ζ (Pkcζ), and the transcription factor zinc finger and BTB domain-containing protein 20 (Zbtb20) [75]. Myo5a and Pkcζ are particularly noteworthy, with Myo5a implicated in the intracellular trafficking of melanosomes [84,85,86] and Pkcζ associated with melanocyte dendrite formation [87].

Furthermore, the Mitfmi-vit allele in the vitiligo substrain of C57BL/6J mice causes late-onset pigment dispersion and increased eye size, potentially due to elevated IOP [72]. This allele disrupts the regulation of several genes including Tyrp1, Dct, Gpnmb, Lyst, Myo5a, and PKCζ, highlighting Mitf’s central role in regulating melanocyte identity and function [64,88]. These findings collectively advance our understanding of the genetic underpinnings of iris pigment dispersion and offer insights into potential therapeutic targets for PDS and related disorders.

### 5.2. Canine Model

In the context of animal models relevant to PDS/PG, canine ocular melanosis (COM) presents a unique case. COM exhibits certain phenotypic overlaps with PDS/PG, such as loss of pigment from the posterior iris leading to transillumination defects, abnormal pigment accumulation in the TM, and elevated IOP culminating in glaucoma in affected canines. However, COM also manifests a spectrum of additional pigmentary and pathogenic phenotypes, including iris root thickening, uveal melanocytic neoplasms, extensive scleral/episcleral pigment plaques, fundus pigmentation, corneal edema, and anterior uveitis [89,90,91]. Genetic analysis in Cairn terriers, assuming an autosomal dominant inheritance pattern, has excluded genes implicated in the DBA/2J mouse model as causative factors in COM [92].

### 5.3. Zebrafish Model

In the emerging zebrafish models, disruption of the *PMEL* homologue, a gene crucial for pigmentation and ocular health, was achieved using CRISPR-Cas9 technology. The targeted mutation, pmelaua5022, involved an 11-base-pair deletion in the C-terminal region of the zebrafish pmela gene, resulting in a frameshift that eliminated 17 amino acids from the pmela protein’s C-terminus. This region is highly conserved across vertebrates, including humans. The mutation led to a significant reduction in pmela transcripts, presumably due to nonsense-mediated decay, with levels 20 times lower in mutant larvae compared to the wild type. The pmelaua5022 homozygous mutant larvae exhibited noticeably reduced global pigmentation and developed ocular abnormalities by 8 days post-fertilization, including more spherical eyes, enlarged anterior segments, and possible indications of high IOP [26,37].

Collectively, animal studies underscore the central role of dysregulated melanin synthesis, compromised melanosome integrity, and impaired melanocyte health in PDS pathogenesis. The identification of multiple genes, each influencing similar biological processes and associated with iris pigment dispersion in mice, is noteworthy. However, there has been no direct association between these genes and PDS/PG in humans. It is crucial to recognize that, although glaucoma develops in both DBA/2J and YBR/EiJ mouse strains, emerging evidence suggests that glaucomatous optic neuropathy might be linked to an underlying neurodegeneration, which is independent of the Tyrp1 alleles [56,93]. This distinction between iris pigment dispersion and the glaucomatous phenotypes observed in these models presents a limitation in their applicability to human PG. Nonetheless, these discrepancies offer valuable insights and contrast with findings from human research in this field. Additionally, the findings in zebrafish support the hypothesis that *PMEL* mutations can induce PDS/PG-like phenotypes, illustrating the zebrafish as an effective and promising model system for studying these conditions. 

## 6. Insights into Disease Pathogenic Pathways

The pathogenesis of PDS and PG involves a complex interplay of genetic and environmental factors. Genetic studies have identified several genes, indicating a significant hereditary component. Notably, genes like *PMEL*, which play a crucial role in melanosome function, melanin synthesis, and storage, have been implicated. Animal models highlight the importance of genes involved in melanin synthesis and melanosome integrity in the pathogenesis of PDS/PG.

Based on our review, we compiled a summary table (Table 3) detailing genes linked to PDS/PG, pathogenic pathways, and evidence levels from human genetics and animal models for PDS/PG. Significantly, *PMEL* stands out as the sole gene substantiated by robust evidence from both human genetics and animal model studies. In contrast, the other genes associated with PDS/PG lack conclusive support from either human genetic research or animal model investigations. This disparity highlights the need for more focused studies in these areas to better understand the genetic underpinnings of PDS/PG. 

Furthermore, we performed a protein–protein interaction and gene enrichment analysis targeting genes associated with PDS/PG (Table 3 and Figure 2). Experimental data and bioinformatics predictions demonstrated significant interactions among DCT, GPNMB, MITF, PMEL, TYR, and TYRP1. These interactions are crucial in a series of biological processes including melanin biosynthesis from tyrosine, developmental pigmentation, melanosome organization, and overall pigmentation, indicating a synergistic role of these genes in the etiology of PDS/PG. Contrastingly, CPAMD8, GSAP, and GRM5 did not exhibit significant interactions with the core PDS/PG-associated genes. This indicates the potential involvement of these genes in PDS/PG via alternative biological pathways or mechanisms, which warrant further investigations.

## 7. Conclusions and Future Directions

In this systematic review, we have provided a comprehensive update on the genetic advancements in PDS/PG, underscoring significant recent progress. These advancements not only enrich our current understanding but also pave the way for future discoveries in PDS/PG genetics. A notable development is the identification of the *PMEL* gene and intriguing insights from GWAS and exome-based analyses, which have been instrumental in elucidating the genetic landscape of PDS/PG.

The functional relevance of identified genes remains an area requiring deeper exploration. Understanding the biological implications and mechanisms of these genes in the context of PDS/PG is essential for translating genetic findings into clinical applications. Lastly, the development of new, more relevant, and efficient model systems is critical for the study of PDS/PG-related genes and variants. Such model systems would not only facilitate a more accurate mimicry of human pathophysiology but also enable the efficient testing of hypotheses derived from genetic studies. In summary, while substantial strides have been made in understanding the genetics of PDS/PG, our review identifies key gaps and outlines future directions necessary for further advancing this crucial field of ocular genetics.

## Figures and Tables

**Figure 1 genes-15-00142-f001:**
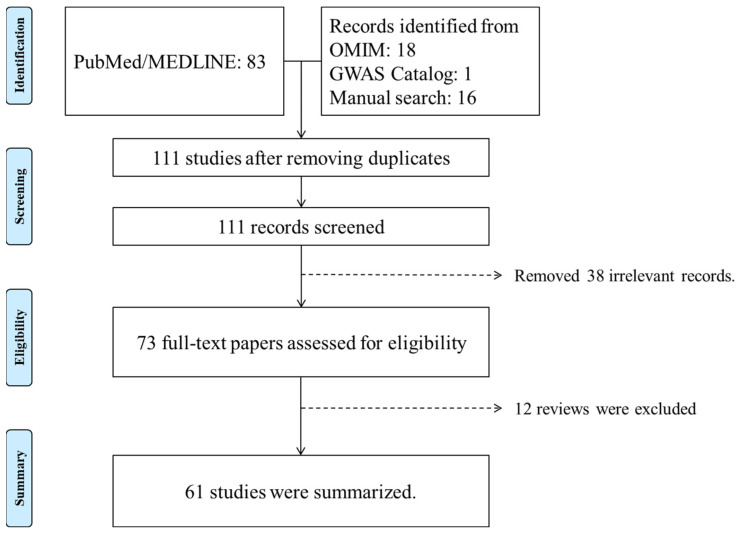
Pigment dispersion syndrome and pigmentary glaucoma genetics literature search and results of literature screening.

**Figure 2 genes-15-00142-f002:**
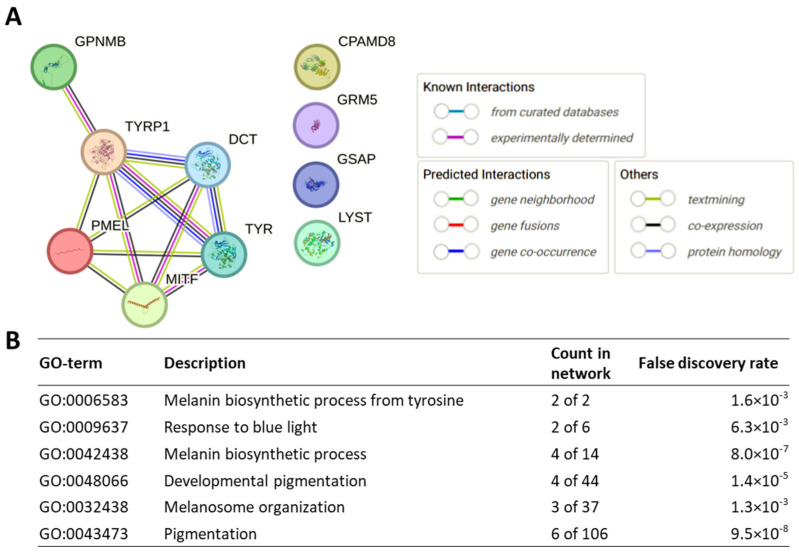
Protein–protein interaction and gene enrichment analysis targeting genes associated with PDS/PG. (**A**) Protein–protein interaction diagram generated based on experimental data and bioinformatics predictions. (**B**) Gene enrichment analysis identifying biological pathways highlighted by the identified genes.

**Table 1 genes-15-00142-t001:** PDS/PG genetic linkage studies and reported genetic loci.

First Author	Year	Phenotype	Number of Families	Method of Genetic Testing	Genetic Loci Identified	Gene Identified	Ancestry of Study Population	Ref.
Wagner, S. H.	2005	PG	4	Microsatellite repeat markers	18q21	Not specified	Caucasian	[29]
Andersen, J. S.	1997	PDS	4	Genome screen using microsatellite repeat markers	7q35-q36	Not specified	White (Irish and mixed western European descent)	[20]

PDS, Pigment Dispersion Syndrome; PG, Pigmentary Glaucoma.

**Table 2 genes-15-00142-t002:** Studies identifying rare variant contributions to PDS/PG.

First Author	Year	Phenotype	Study Population	Sample Size	Method of Genetic Testing	Study Population	Gene	Ref.
Tan, J.	2022	PDS/PG	Family & unrelated samples	2 pedigrees; 38 sporadic patients	WES	Chinese	*CPAMD8*	[24]
van der Heide, C.	2021	PG	Unrelated samples	415 cases; 1645 controls	WES	Caucasian	*MRAP*	[25]
Lahola-Chomiak, A. A.	2019	PG	Family	2 pedigrees; 394 in cohorts	WES; Targeted screening	Caucasian	*PMEL*	[26]

PDS, Pigment Dispersion Syndrome; PG, Pigmentary Glaucoma; WES, whole exome sequencing.

**Table 3 genes-15-00142-t003:** Summary of evidence level and biological insights based on PDS/PG genetic studies.

Candidate Genes	Potential Pathogenesis Pathways	Evidence Level (Y/N)	
		Human Genetics	Animal Models
*PMEL*	Melanosome function, melanin synthesis, storage	Yes	Yes (Zebrafish, Mouse)
*GSAP*	Pigmentation processes	Yes (GWAS)	No
*GRM5/TYR*	Ocular pigmentation, retinal detachment	Yes (GWAS)	No
*CPAMD8*	Ocular phenotypes, anterior segment dysgenesis	Yes	No
*TYRP1*	Iris pigment dispersion, melanin synthesis	No	Yes (Mouse)
*GPNMB*	Melanosome structural integrity, pigmentation	No	Yes (Mouse)
*LYST*	Melanosome function, melanin synthesis intermediates	No	Yes (Mouse)
*MITF*	Regulation of melanocyte identity and function	No	Yes (Mouse)
*DCT*	Plays a critical role in melanin synthesis	No	Yes (Mouse)
*TBK1*	Distinct genetic basis from other glaucoma types	No	No
*LOXL1*	Influence on expression and risk of PDS/PG	No	No
*MRAP*	Possible link to PDS, role in iris and ocular tissues	No	No

## Data Availability

We used publicly available software for the analyses and provided a list of the specific programs used in the methods section.

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
