# Peer review of "Genetic Basis of Pigment Dispersion Syndrome and Pigmentary Glaucoma: An Update and Functional Insights"

_genes, 2024, doi:10.3390/genes15020142_

Round 1

Reviewer 1 Report

Comments and Suggestions for Authors

This article provides a comprehensive review on the status of PDS and PG genetics from the recent genetic technological development including genome-wide association studies, whole exome sequencing, and whole genome sequencing. The article also discussed the relationship of PDS with other known genetic disorders. At the end, the article provided more discussion around different animal models in relation to disease pathogenesis. This article will be a good addition to the available literature to the field of PDS/PG genetics. This manuscript could be improved by adding a few tables for each major section and a conclusive graphic summary about the current understanding. These new tables and the summary figures should provide readers with more advanced understanding of the relevant topic.

Author Response

Dear Reviewer,

Thank you for your comments on our manuscript about PDS and PG genetics. We are grateful for your appreciation of our review.

In response to your valuable suggestions, we have made significant enhancements to the manuscript. We have included two additional figures and three summary tables, each correlating with the major sections of the article. Additionally, a supplementary table has been added for better understanding. These additions are designed to provide a more advanced and graphical summary of the current understanding in the field of PDS/PG genetics.

We have conducted a thorough revision of the entire manuscript. All changes and additions have been highlighted in yellow for ease of review. We believe these updates will greatly improve the manuscript and provide readers with a more comprehensive understanding of the topic.

Thank you again for your constructive feedback and for the opportunity to enhance our work.

Reviewer 1

This article provides a comprehensive review on the status of PDS and PG genetics from the recent genetic technological development including genome-wide association studies, whole exome sequencing, and whole genome sequencing. The article also discussed the relationship of PDS with other known genetic disorders. At the end, the article provided more discussion around different animal models in relation to disease pathogenesis. This article will be a good addition to the available literature to the field of PDS/PG genetics. This manuscript could be improved by adding a few tables for each major section and a conclusive graphic summary about the current understanding. These new tables and the summary figures should provide readers with more advanced understanding of the relevant topic.

Authors’ responses: Thank you for your insightful comments on our manuscript.

We have made significant enhancements to the manuscript, including two additional figures, three summary tables, a supplementary table.

A thorough revision of the entire manuscript has been done. All changes and additions have been highlighted in yellow for ease of review.

Figure 1. Pigment dispersion syndrome and pigmentary glaucoma genetics literature search and results of literature screening

Table 1. PDS/PG genetic linkage studies and reported genetic loci

First Author

Year

Phenotype

Number of Families

Method of Genetic Testing

Genetic Loci Identified

Gene Identified

Ancestry of Study Population

Ref

Wagner, S. H.

2005

PG

4

Microsatellite repeat markers

18q21

Not specified

Caucasian

[1]

Andersen, J. S.

1997

PDS

4

Genome screen using microsatellite repeat markers

7q35-q36

Not specified

White (Irish and mixed western European descent)

[2]

PDS, Pigment Dispersion Syndrome; PG, Pigmentary Glaucoma

Table S2. Genome-wide association studies and candidate gene studies of pigment dispersion syndrome and pigmentary glaucoma

First Author

Year

Phenotype

Sample Size

Genetic analysis

Study Population

Genes

Significant
association

Ref

Genome-wide association study

1

Simcoe, M. J.

2022

PDS/PG

Cases (574), controls (52,627)

GWAS

European

GSAP, GRM5/TYR

Yes

[3]

Candidate gene study

1

Fingert, J. H.

2016

PG

PG (209), no controls

qPCR assay, CNV

Caucasian

TBK1

No

[4]

2

Giardina, E.

2014

PDS/PG

Patients (84), controls (200)

GA

Caucasian

LOXL1

No

[5]

3

Wolf, C.

2010

PG

PG (88), controls (280)

GA

Caucasian

LOXL1

No

[6]

4

Rao, K. N.

2008

PDS/PG

PG (44), PDS (34), controls (108)

GA

Caucasian

LOXL1

No

[7]

5

Lynch, S.

2002

PG

PG probands from 4 families

Sequencing of TYRP1 gene

Caucasian

TYRP1

No

[8]

6

Jaksic, V.

2010

PG

1 Patient

Case report

Caucasian

MTHFR C677T

na

[9]

CNV, copy number variants; GA, genetic case-control association study; GWAS, Genome-Wide Association Meta-Analysis; na, not applicable; PDS, Pigment Dispersion Syndrome; PG, Pigmentary Glaucoma; qPCR, Quantitative Polymerase Chain Reaction

Table 2. Studies identifying rare variant contributions to PDS/PG

First Author

Year

Phenotype

Study Population

Sample Size

Method of Genetic Testing

 Study Population

Gene

Ref

Tan, J.

2022

PDS/PG

Family

2 pedigrees; 38 sporadic patients

WES

Chinese

CPAMD8

[10]

van der Heide, C.

2021

PG

Unrelated samples

415 cases; 1645 controls

WES

Caucasian

MRAP

[11]

Lahola-Chomiak, A. A.

2019

PG

Family

2 pedigrees; 394 in cohorts

WES; Targeted screening

Caucasian

PMEL

[12]

PDS, Pigment Dispersion Syndrome; PG, Pigmentary Glaucoma; Whole exome sequencing

Table 3. Summary of evidence level and biological insights based on PDS/PG genetic studies    

Candidate genes

Potential Pathogenesis Pathways

Evidence level (Y/N)

 Human Genetics

Animal Models

PMEL

Melanosome function, melanin synthesis, storage

Yes

Yes (Zebrafish, Mouse)

GSAP

Pigmentation processes

Yes (GWAS)

No

GRM5/TYR

Ocular pigmentation, retinal detachment

Yes (GWAS)

No

CPAMD8

Ocular phenotypes, anterior segment dysgenesis

Yes

No

TYRP1

Iris pigment dispersion, melanin synthesis

No

Yes (Mouse)

GPNMB

Melanosome structural integrity, pigmentation

No

Yes (Mouse)

LYST

Melanosome function, melanin synthesis intermediates

No

Yes (Mouse)

MITF

Regulation of melanocyte identity and function

No

Yes (Mouse)

DCT

Plays a critical role in melanin synthesis

No

Yes (Mouse)

TBK1

Distinct genetic basis from other glaucoma types

No

No

LOXL1

Influence on expression and risk of PDS/PG

No

No

MRAP

Possible link to PDS, role in iris and ocular tissues

No

No

Figure 2. Protein-protein interaction and gene enrichment analysis targeting genes associated with PDS/PG. (A) Protein-protein interaction diagram generated based on experimental data and bioinformatics predictions. (B) Gene enrichment analysis identifying biological pathways highlighted by the identified genes.

Reference

  1. Wagner, S.H.; DelBono, E.; Greenfield, D.S.; Parrish, R.K.; Haines, J.L.; Wiggs, J.L. A Second Locus for Pigment Dispersion Syndrome Maps to Chromosome 18q21. In Proceedings of the ARVO Annual Meeting, Fort Lauderdale, FL, 2005.
  2. Andersen, J.S.; Pralea, A.M.; DelBono, E.A.; Haines, J.L.; Gorin, M.B.; Schuman, J.S.; Mattox, C.G.; Wiggs, J.L. A gene responsible for the pigment dispersion syndrome maps to chromosome 7q35-q36. Archives of ophthalmology (Chicago, Ill. : 1960) 1997, 115, 384-388, doi:10.1001/archopht.1997.01100150386012.
  3. Simcoe, M.J.; Shah, A.; Fan, B.; Choquet, H.; Weisschuh, N.; Waseem, N.H.; Jiang, C.; Melles, R.B.; Ritch, R.; Mahroo, O.A.; et al. Genome-Wide Association Study Identifies Two Common Loci Associated with Pigment Dispersion Syndrome/Pigmentary Glaucoma and Implicates Myopia in its Development. Ophthalmology 2022, 129, 626-636, doi:10.1016/j.ophtha.2022.01.005.
  4. Fingert, J.H.; Robin, A.L.; Scheetz, T.E.; Kwon, Y.H.; Liebmann, J.M.; Ritch, R.; Alward, W.L. Tank-Binding Kinase 1 (TBK1) Gene and Open-Angle Glaucomas (An American Ophthalmological Society Thesis). Transactions of the American Ophthalmological Society 2016, 114, T6.
  5. Giardina, E.; Oddone, F.; Lepre, T.; Centofanti, M.; Peconi, C.; Tanga, L.; Quaranta, L.; Frezzotti, P.; Novelli, G.; Manni, G. Common sequence variants in the LOXL1 gene in pigment dispersion syndrome and pigmentary glaucoma. BMC ophthalmology 2014, 14, 52, doi:10.1186/1471-2415-14-52.
  6. Wolf, C.; Gramer, E.; Müller-Myhsok, B.; Pasutto, F.; Gramer, G.; Wissinger, B.; Weisschuh, N. Lysyl oxidase-like 1 gene polymorphisms in German patients with normal tension glaucoma, pigmentary glaucoma and exfoliation glaucoma. Journal of glaucoma 2010, 19, 136-141, doi:10.1097/IJG.0b013e31819f9330.
  7. Rao, K.N.; Ritch, R.; Dorairaj, S.K.; Kaur, I.; Liebmann, J.M.; Thomas, R.; Chakrabarti, S. Exfoliation syndrome and exfoliation glaucoma-associated LOXL1 variations are not involved in pigment dispersion syndrome and pigmentary glaucoma. Molecular vision 2008, 14, 1254-1262.
  8. Lynch, S.; Yanagi, G.; DelBono, E.; Wiggs, J.L. DNA sequence variants in the tyrosinase-related protein 1 (TYRP1) gene are not associated with human pigmentary glaucoma. Molecular vision 2002, 8, 127-129.
  9. Jaksic, V.; Markovic, V.; Milenkovic, S.; Stefanovic, I.; Jakovic, N.; Knezevic, M. MTHFR C677T homozygous mutation in a patient with pigmentary glaucoma and central retinal vein occlusion. Ophthalmic research 2010, 43, 193-196, doi:10.1159/000272023.
  10. Tan, J.; Zeng, L.; Wang, Y.; Liu, G.; Huang, L.; Chen, D.; Wang, X.; Fan, N.; He, Y.; Liu, X. Compound Heterozygous Variants of the CPAMD8 Gene Co-Segregating in Two Chinese Pedigrees With Pigment Dispersion Syndrome/Pigmentary Glaucoma. Frontiers in genetics 2022, 13, 845081, doi:10.3389/fgene.2022.845081.
  11. van der Heide, C.; Goar, W.; Meyer, K.J.; Alward, W.L.M.; Boese, E.A.; Sears, N.C.; Roos, B.R.; Kwon, Y.H.; DeLuca, A.P.; Siggs, O.M.; et al. Exome-based investigation of the genetic basis of human pigmentary glaucoma. BMC genomics 2021, 22, 477, doi:10.1186/s12864-021-07782-0.
  12. Lahola-Chomiak, A.A.; Footz, T.; Nguyen-Phuoc, K.; Neil, G.J.; Fan, B.; Allen, K.F.; Greenfield, D.S.; Parrish, R.K.; Linkroum, K.; Pasquale, L.R.; et al. Non-Synonymous variants in premelanosome protein (PMEL) cause ocular pigment dispersion and pigmentary glaucoma. Human molecular genetics 2019, 28, 1298-1311, doi:10.1093/hmg/ddy429.

Reviewer 2 Report

Comments and Suggestions for Authors

The authors have submitted a review with the title Genetic Basis of Pigment Dispersion Syndrome and Pigmentary Glaucoma: An Update and Functional Insights to MDPI Genes. The article, written by two authors, consists of an unstructured abstract, 7 key words, 6 subsections, editorial statements, 101 references and a supplemental table and figure with caption. The authors are advised to follow the guidelines of the journal and respect standards of good research practice. 

The present review article summarises and updates on the current literature on the well-researched topic of Pigment Dispersion Syndrome and Pigmentary Glaucoma. The review is based on a systematic literature search with specific relevance to the genetic background of the condition. Highlighting developments and recent advances in this field is relevant due to the basic science (genetic) research in the field and the progress in advanced therapeutics which in the future could hopefully allow to treat the condition with a more targeted approach. 

The review is well written and comprehensively discusses the topic including links and contexts to other ophthalmic or systemic disorders from a genetic/scientific perspective. 

In-text corrections: 

L150-152 & L160-161: The two sentences are repetitive, the statement should be made only once. 

L416: remove ")" after PDS. 

L445: Please specify the specific contribution that the acknowledged person has made. Please briefly explain why the person is not listed as an author. 

Comments on the Quality of English Language

Solid English, very minor mistakes in grammar

Author Response

Dear Reviewer,

Thank you for your valuable feedback on our manuscript titled "Genetic Basis of Pigment Dispersion Syndrome and Pigmentary Glaucoma: An Update and Functional Insights".

We have thoroughly reviewed and addressed each of the comments you provided. In line with your advice, we have ensured our submission adheres to the guidelines of the journal and upholds the standards of good research practice.

Additionally, we have undertaken a meticulous revision of the entire manuscript. To facilitate your review, all changes have been clearly highlighted in yellow. This revision not only incorporates your suggestions but also reinforces the clarity and depth of our review.

We appreciate your constructive feedback.

Reviewer 2

The authors have submitted a review with the title Genetic Basis of Pigment Dispersion Syndrome and Pigmentary Glaucoma: An Update and Functional Insights to MDPI Genes. The article, written by two authors, consists of an unstructured abstract, 7 key words, 6 subsections, editorial statements, 101 references and a supplemental table and figure with caption. The authors are advised to follow the guidelines of the journal and respect standards of good research practice.

The present review article summarizes and updates on the current literature on the well-researched topic of Pigment Dispersion Syndrome and Pigmentary Glaucoma. The review is based on a systematic literature search with specific relevance to the genetic background of the condition. Highlighting developments and recent advances in this field is relevant due to the basic science (genetic) research in the field and the progress in advanced therapeutics which in the future could hopefully allow to treat the condition with a more targeted approach.

The review is well written and comprehensively discusses the topic including links and contexts to other ophthalmic or systemic disorders from a genetic/scientific perspective.

In-text corrections:

L150-152 & L160-161: The two sentences are repetitive, the statement should be made only once.

Authors’ response: Thank you for your feedback. We appreciate your attention to detail. We have revised the text to eliminate the repetition and ensure that the statement is clear and concise, conveyed effectively in a single sentence.

“Studies exploring the association between the LOXL1 gene polymorphisms and PDS/PG have yielded inconclusive results mainly due to insufficient statistical power in each study, necessitating a comprehensive meta-analysis.”

L416: remove ")" after PDS.

Authors’ response: Thank you for pointing out the typographical error. I have promptly removed the extraneous ")" after "PDS" to correct the text. Your attention to detail is greatly appreciated and helps in maintaining the accuracy and clarity of our work.

L445: Please specify the specific contribution that the acknowledged person has made. Please briefly explain why the person is not listed as an author.

Authors’ response: Thank you for your comment. We have added detailed authors’ contribution to the m/s.

“Author Contributions:

Conception: SSR

Data curation: SSR, XTY

Funding acquisition: SSR

Investigation: SSR, JLW

Methodology: SSR

Project administration: SSR

Resources: SSR, XTY

Supervision: JLW

Writing - original draft: SSR, XTY

Writing - review & editing: SSR, JLW”

Round 2

Reviewer 1 Report

Comments and Suggestions for Authors

The authors have added several tables and figures to improve the overall quality of the review article. The review is very comprehensive and easy to follow. There are no additional comments.